# Light-to-Heat Converting ECM-Mimetic Nanofiber Scaffolds for Neuronal Differentiation and Neurite Outgrowth Guidance

**DOI:** 10.3390/nano12132166

**Published:** 2022-06-23

**Authors:** Olga Yu. Antonova, Olga Yu. Kochetkova, Igor L. Kanev

**Affiliations:** Institute of Theoretical and Experimental Biophysics, Russian Academy of Sciences, 142290 Pushchino, Moscow region, Russia; o.y.kochetkova@gmail.com (O.Y.K.); 4kanev@gmail.com (I.L.K.)

**Keywords:** light-to-heat converting nanoparticles, nanofibers, fibrous scaffolds, near infrared irradiation, photothermal stimulation, neuronal differentiation, neurites growth guidance

## Abstract

The topological cues of fibrous scaffolds (in particular extracellular matrix (ECM)-mimetic nanofibers) have already proven to be a powerful tool for influencing neuronal morphology and behavior. Remote photothermal optical treatment provides additional opportunities for neuronal activity regulation. A combination of these approaches can provide “smart” 3D scaffolds for efficient axon guidance and neurite growth. In this study we propose two alternative approaches for obtaining biocompatible photothermal scaffolds: surface coating of nylon nanofibers with light-to-heat converting nanoparticles and nanoparticle incorporation inside the fibers. We have determined photoconversion efficiency of fibrous nanomaterials under near infrared (NIR) irradiation, as well as biocompatible photothermal treatment parameters. We also measured photo-induced intracellular heating upon contact of cells with a plasmonic surface. In the absence of NIR stimulation, our fibrous scaffolds with a fiber diameter of 100 nm induced an increase in the proportion of β3-tubulin positive cells, while thermal stimulation of neuroblastoma cells on nanoparticles-decorated scaffolds enhanced neurite outgrowth and promoted neuronal maturation. We demonstrate that contact guidance decorated fibers can stimulate directional growth of processes of differentiated neural cells. We studied the impact of nanoparticles on the surface of ECM-mimetic scaffolds on neurite elongation and axonal branching of rat hippocampal neurons, both as topographic cues and as local heat sources. We show that decorating the surface of nanofibers with nanoparticles does not affect the orientation of neurites, but leads to strong branching, an increase in the number of neurites per cell, and neurite elongation, which is independent of NIR stimulation. The effect of photothermal stimulation is most pronounced when cultivating neurons on nanofibers with incorporated nanoparticles, as compared to nanoparticle-coated fibers. The resulting light-to-heat converting 3D materials can be used as tools for controlled photothermal neuromodulation and as “smart” materials for reconstructive neurosurgery.

## 1. Introduction

Application of fibrous nanomaterials in areas of social and economic importance, such as regenerative medicine and biotechnology, has experienced significant progress in recent years [1,2,3,4,5].

Due to specific properties associated with their structure, use of nanofibers is considered a promising approach to control behavior and functions of biological systems, especially the nervous system. The ability to control and direct the growth of axons is the most important characteristic of a nanomaterial used for the regeneration of damaged nerve tissue. Surface nanotopology is one of the powerful factors that allows accelerating and directing the growth of nerve cells [6].

The electrospinning technique is considered one of the most effective methods to prepare nanofibers with efficient control of patterning and assembly [7]. An important feature of this method is the ability to obtain fibrous nanomaterials consisting of fibers oriented in one direction [8,9,10]. Natural extracellular matrix (ECM) architectures and topographies affect neural cell properties and functions, such as cell migration, differentiation, and morphology through contact guidance, carried out by mechanical (topology) and biochemical cues [11]. The ability to mimic the architecture of natural ECM, combined with the large specific area of fibrous material, allow nanocomposite scaffolds to encourage neuron growth [12,13,14]. When designing ECM-mimetic scaffolds, it is necessary to take into account the dimensions of the structural components of the neural ECM; e.g., collagen fibrils of individual endoneurium have an average diameter of 41.31 ± 4.2 nm and 30–60 nm in perineurium layer [15]. In addition to the ability to control the morphology of the resulting nanomaterial, electrospinning makes it easy to incorporate other substances, such as conductive polymers, nanotubes, nanowires, nanoparticles, etc., that can act on the interacting cells as coadjuvants or stimulants [16,17]. Thus, Tiwari et al. [18] presented a fibrous site-specific drug delivery platform functionalized with pH and NIR-responsive polypyrrol developed by electrospinning technique combined with subsequent processing. Electrospinning also provides multiple ways to produce composite materials consisting of fibers decorated with nanoparticles [7]. Several studies have demonstrated a tremendous potential of nanoparticles immobilized on a substrate to enhance neuronal proliferation, axon growth, neuronal cell adhesion, as well as confer neuroprotective effects [12,19]. In addition, the large surface of the nanomaterial provides ample opportunities for nanoparticle decoration or conjugation with other molecules to enhance the effect on nerve cells [20,21]. In addition to creating nanotopological patterns on the surface of the substrate, attachment of thermoplasmonic nanoparticles (TNPs) makes it possible to use another powerful stimulus to control the growth of neurons—local thermal stimulation. Thermal exposure is a promising method for modulating cellular functions, providing controlled inhibition or stimulation of neuronal activity by varying the intensity of laser radiation, as well as the optical characteristics (photothermal conversion efficiency) of the nanomaterial [22,23,24,25].

Near-infrared (NIR) radiation, due to its capacity for deep penetration into live tissues, provides efficient means for the remote control of nanoparticles in vivo through local thermal activation. A number of recent works have been focused on the means to control cellular activity on composite photothermal materials obtained by various methods: hydrogels, 3d printing, electrospinning, etc. [26]. Thus, photothermal stimulation using nanoparticles that convert NIR radiation into heat has shown promising opportunities for killing skin tumor cells, healing tissues [27,28], and also stimulation and inhibition of neuronal activity [29,30,31,32]. At the same time, TNPs are able to provide spatially and temporally accurate local heat delivery: it was shown that by stimulating individual cells, it is possible to control the activity of the neighboring neurons, and the neural network as a whole [33,34,35]. Application of gold nanostructures for neurochemical sensing, neuromodulation, neuroimaging, neurotherapy, tissue engineering, and neural regeneration is literally becoming the “gold standard” [36]. However, their use for photothermal treatment is associated with a number of difficulties: complex multi-stage synthesis and the need for stabilization, strong dependence of the plasmon resonance peak on the geometry of the nanoparticle, low efficiency of photoconversion, low photostability, low rate of clearance from the body [37,38]. Recently CuS nanoparticles emerged as promising photothermal agents and a viable alternative to those based on gold. The advantages of CuS nanoparticles include their low cost, high photostability, and low cytotoxicity, due to Cu being a physiological metal (unlike Au); Cu ions act as a cofactor for a number of enzymes, enhance angiogenesis, and activate signaling pathways involved in regeneration [39]. Absorption of CuS NPs does not depend on the dielectric constant of the surrounding medium and is not significantly affected by nanoparticle morphology [40]. Application of biomineral synthesis based on protein nanoreactors makes it possible to simplify the procedure for obtaining CuS nanocrystals [38]. In addition, the protein component of such nanoparticles contributes to colloidal stability, and the presence of functional groups can be used for immobilization by chemical cross-linking, chemical modification, [41] etc. In summary, development of nanomaterials that combine the advantages of ECM-mimetic topology, thermo-induced stimulation of neuronal differentiation processes, and are capable of accelerating the directional growth of axons, represents a particularly promising direction in technological developments for neurotherapy and tissue engineering.

Previously, we have described neuron growth on electrospun scaffolds of nylon-4,6 aligned ultrathin (AU) fibers with an average diameter of 60 nm, which closely match the diameter of collagen nanofibers of neural ECM. This biocompatible nanomaterial has been shown to stimulate directed accelerating neurite elongation, improved synaptogenesis, and formation of connections between hippocampal neurons [42]. In this work, we present a hybrid biocompatible nanomaterial consisting of oriented polymeric nanofibers decorated with TNPs using alternative methods: internal impregnation during electrospinning process and chemical immobilization on the surface. The resulting hybrid materials demonstrate photothermal activity when exposed to NIR irradiation (λ = 808 nm). Local increase in intracellular temperature is registered when cells are exposed to hybrid scaffolds under NIR radiation. The surface topology of the resulting nanomaterial and photoinduced heating exert a synergistic effect on the differentiation of neuroblastoma cells. We also study the potential of local photothermal stimulation for controlling axon guidance and neurite elongation on the surface of a fibrous nanomaterial.

## 2. Materials and Methods

### 2.1. Chemicals

Antibodies: anti-β3-tubulin antibody [2G10] (Alexa Fluor 488) from Abcam (Cambridge, UK); anti-GFAP antibody (Alexa Fluor 594) from Biolegend, (San Diego, CA, USA). All-trans-retinoic acid, B27 supplement, bovine serum albumin (BSA), calcein-acetoxymethyl (calcein-AM), Dil [DilC18(3)], Fluo-4 AM, Pluronic F127 from Invitrogen (Carlsbad, CA, USA); Dulbecco’s modified Eagle’s medium (DMEM) /F12 from Paneco (Moscow, Russia), fetal bovine serum from Gibco (Gaithersburg, MD, USA), 4% formaldehyde in PBS, glycine, glutamine from Paneco (Moscow, Russia), 25% glutaraldehyde, 98% formic acid, hexamethyldisilazane, Neurobasal-A medium from Gibco (Gaithersburg, MD, USA), nylon-4,6, penicillin-streptomycin mixed solution, phalloidin-iFluor 555 from Abcam, PBS from Paneco (Moscow, Russia), 25 kDa polyethyleneimine (PEI), poly-D-lysine, propidium iodide (PI), sodium dodecyl sulfate (SDS), 25% Triton X-100, trypsin, Tween 20, Rodamine B (Rhod B). Cell culture 24-well plates, tissue culture flasks from SPL Lifesciences (Pocheon-si, South Korea). All other chemicals were purchased from Sigma-Aldrich (Saint Louis, MO, USA), unless otherwise indicated.

### 2.2. Animals

All animal studies were approved by the Animal Ethics Committee of the Institute of Theoretical and Experimental Biophysics RAS.

### 2.3. Preparation of Nanofibrous Scaffolds

Aligned ultrathin (AU) nylon fibers were prepared as described earlier [43]. The polymer solution (12% *w*/*v*) was prepared by dissolving nylon-4,6 (Sigma-Aldrich, St. Louis, MO, USA) in formic acid by overnight rotational mixing. To prepare CB-NPs containing fibers freshly prepared nanoparticles were dispersed in formic acid before nylon addition. Nanomats from aligned fibers were made by electrospinning polymer solution onto a grounded wire drum with a diameter of 108 mm and a 13 mm distance between adjacent wires. The drum was fixed on a plastic frame 10 cm from the needle, and the rotation speed was 80 rpm. Nylon solution was pumped through a 0.4 mm steel needle with a volume rate of 0.5 μL/min. The electrospray current was 250−350 nA. Every 10 min, the electrospray was stopped and the formed mats were treated with an air ionizer to electrically neutralize the fresh-formed fibers and the plastic frame. Prepared electrospun fibers were transferred to sterile glass plates (9 × 11 × 1 mm) and then glued at the edges with a polystyrene solution in toluene. Nanofiber samples were characterized using a SmartSPM-1000 atomic force microscope (AFM, AIST-NT Co, Moscow, Russia) in tapping mode and a scanning electron microscope (SEM) at 2 kV.

### 2.4. Synthesis of TNPs

CuS-BSA nanoparticles were synthesized as described by Zhang et al. [39,44]. After synthesis, the nanoparticles were centrifuged at 18,000× *g* rpm (60,000× *g*). To modify the nanofibers by chemical immobilization, the precipitate was dissolved in a 0.1 M HEPES solution to prevent particle aggregation. Morphology and microstructure of the CuS-BSA nanoparticles were examined using a Supra 50 VP LEO high-resolution scanning electron microscope with an INCA microanalysis system INCA Energy+ Oxford (LEO Carl Zeiss SMT Ltd., Oberkochen, Germany) at an accelerating voltage of 21 kV. The UV-Vis-NIR absorption spectra of nanoparticles were recorded on a CARY 5000 UV-Vis-NIR spectrophotometer (Agilent Technologies, Santa Clara, CA, USA).

### 2.5. Preparation of TNPs-Modified Nanofibers

Aligned ultrathin nylon fibers with incorporated nanoparticles (Type AU-In) were prepared by adding freshly prepared CB-NPs (1.5% *w*/*v*) to the nylon solution and overnight rotational mixing. The conditions for electrospinning and electrospraying were the same as for the AU fibers control. Fibers coated with nanoparticles (Type AU-Coat) were prepared by chemical immobilization of freshly prepared CB-NPs onto AU fibers glued to a glass plate. AU fibers were pretreated in plasma for 5 s, then the oxidized nylon surface was coated with branched 25 kDa PEI, then CB-NPs were crosslinked to the PEI layer with glutaraldehyde. The scheme of immobilization and images of composite scaffolds are presented in the Appendix A. Scanning electron microscopy (SEM) images and energy-dispersive X-ray spectra (EDS) were recorded on a Tescan Amber (Tescan, Czech) microscope operated at 20 kV and equipped with EDS Ultim Max 100 detector (Oxford Instruments Inc., Abingdon, UK). The elemental composition analysis and elemental map reconstruction were performed using the Aztec (Oxford Instruments Inc., Abingdon, UK) software in an automatic mode. Sterilization of samples prior to use for cell cultivation was performed as follows: AU and AU-In fibers were treated for 5 s in plasma, and AU-Coat samples were washed with sterile PBS.

### 2.6. Cell Culture

#### 2.6.1. Rat Hippocampal Neurons

Hippocampal neurons were derived from the brain of newborn (1–3 days old) Sprague–Dawley rats according to the protocol [45]. Briefly, the hippocampi were dissected and separated from meninges and surrounding tissue prior to enzymatic digestion with 0.25% *w*/*v* trypsin for 10 min at 37 °C. After pipetting and centrifugation (2000 rpm, 5 min), the resulting cell pellet was resuspended in the hippocampus Neurobasal-A medium and 2% *v*/*v* B27 solution. Afterwards, 5 × 10^4^ cells were placed on glass plates (similar to those used in fibrous scaffolds) coated with 0.1 mg/mL poly-D-lysine, which served as a control, and on nanofiber samples in separate wells of a 24-well plate. Neurons were cultivated at 37 °C and 5% *v*/*v* CO_2_ in a humidified incubator.

#### 2.6.2. Human Neuroblastoma Cells SH-SY5Y

SH-SY5Y cells were purchased from the Russian Cell Culture Collection (Institute of Cytology RAS, Saint Petersburg, Russia). SH-SY5Y were maintained in DMEM/F12 supplemented with 10% heat-inactivated fetal bovine serum and 1% penicillin–streptomycin solution (Proliferation medium). For cell differentiation 24 h after seeding, the Proliferation medium was replaced with the Differentiation medium: DMEM/F12 supplemented with 1% heat-inactivated FBS, 1% penicillin-streptomycin, and 10 μM all-trans-retinoic acid. Further manipulations with cells were performed according to the scheme on Figure 1.

### 2.7. Cell Viability Assay

SH-SY5Y cells were seeded at 7.5 × 10^4^ cells per sample and incubated for 24 h in the Proliferation medium and then incubated for 72 h in the Differentiation medium. After 72 h, the cells were treated with NIR irradiation at different power densities (0.5, 1, 3, and 5 W/cm^2^ for 5 min), with beam diameter of 10 mm for 0.5, 1, 3 W/cm^2^ and 5 mm for 5 W/cm^2^ (Figure 1A). Cells were analyzed 24 h later using a LIVE/DEAD cell viability assay. For 20 min, 1 μM of propidium iodide and 2 μM of calcein-AM were added to the medium. Samples were then fixed with 4% paraformaldehyde (PFA), washed with PBS, and imaged by fluorescence microscopy (Axiovert 200M, Zeiss, Göttingen, Germany) using a 20× objective (numerical aperture (NA) 0.3).

### 2.8. Neuronal Differentiation of SH-SY5Y Cells

SH-SY5Y cells were seeded at 7.5 × 10^4^ cells per sample and incubated for 24 h in the Proliferation medium. Then the Proliferation medium was replaced with the Differentiation medium, and cells were treated with NIR irradiation (808 nm) at different power densities (0.5, and 1 W/cm^2^ for 5 min) with a beam diameter of 10 mm (Figure 1B). The cells were further cultured for 48 h and immunostained as described further. After fixation with 4% PFA for 10 min at RT, the cells were permeabilized in 0.1% Triton X-100 for 5 min and then blocked with the solution containing 1% BSA, 10% serum, 0.3 M glycine in 0.1% PBS-Tween 20 for 1 h. Fixed cells were incubated overnight at 4 °C with fluorescent-labeled (Alexa Fluor 488) anti-βIII Tubulin antibodies. For actin staining, phalloidin-iFluor 555 reagent (1:1000) was added to the antibody solution. Nuclear staining was performed with 1% Hoechst 33258 solution for 10 min. Finally, the cells were mounted on a glass slide and examined by fluorescence microscopy using a 20× air objective. Fluorescent images of fibers stained with Hoechst were acquired with excitation at 405 nm and emission 430–480 nm. Experiments were carried out in three or more replicates. For each replicate three independent measurements were performed, in which at least 100 cells were analyzed. The length and orientation of neurites were analyzed using NeuronJ and OrientationJ plugins for ImageJ software (v. 1.5, National Institutes of Health [NIH]) as described earlier [42].

### 2.9. Morphological Assay of Rat Hippocampal Neuron

Isolated cells were plated on AU fibers and TNPs-modified scaffolds at 5 × 10^4^ cells per sample and incubated for 24 h. Thereafter the cells were exposed to 808 nm NIR radiation at a power density of 1 and 3 W/cm^2^ and cultured for 6 days. The cells were then fixed with 4% PFA solution and stained with anti-β III Tubulin (Alexa Fluor 488) and anti-GFAP antibodies (Alexa Fluor 594) as described above. Images were obtained using a fluorescent microscope with a 20× air objective (NA 0.3). Analysis of neurite elongation was performed using the NeuronJ plugin. To analyze the fraction of neurons in the overall cell population, β3-tubulin-positive cells were counted. Cell viability was performed by Hoechst staining. Cells with aggregated or fragmented chromatin were regarded as apoptotic cells according to [46].

### 2.10. Photothermal Performance

#### 2.10.1. Micro-Scale Measurement of Plasmonic Nanofiber Temperature during NIR Stimulation

Micro-scale temperature change measurements were performed by fluorescence thermal imaging, according to [47]; Rhod B was used as a thermosensitive probe. For the calibration procedure, nanofiber samples were incubated in a 50 μM Rhod B solution for 1 h, then washed with water and dried. Fluorescence intensity was calibrated within the range of temperatures from 20 °C to 45 °C using a custom homemade external heating stage. Fluorescent images were acquired at each temperature point taken. Imaging of dry samples (samples without liquid) was performed using an Axiovert 200 microscope (Carl Zeiss, Göttingen, Germany) with a 63× air objective (NA 0.7). Heating temperature was controlled with a thermocouple. To evaluate the photothermal performance of TNPs-modified nanofibers, samples were irradiated by a 808 nm laser (ATC-C4000-200-AMF-808-3-F400, ATC-Semiconductor Devices, Saint Petersburg, Russia) at a power density of 3.0 W/cm^2^ with a beam diameter of 4 mm for 5 min. Temperature changes of the samples were determined by fluorescent image analysis. Temperature changes of the control glass plate were measured under the same experimental conditions.

#### 2.10.2. Intracellular Temperature Detection

Hippocampal neurons were seeded at 5 × 10^4^ cells per well in a 4-well plate. The photothermal effect of AU fibers and TNPs-modified scaffolds was analyzed in hippocampal neurons cultured in vitro for 4 or more days. Neurons were incubated for 1 h in 1 mL of culture medium containing 50 μM Rhod B per well for and then exposed for 5 min to the 808 nm laser irradiation at the given power density with a beam diameter of 10 mm. Imaging was performed using an Axiovert 200 microscope with a 63× oil objective (NA, 0.7). The fluorescent images of stained cells and nanofibers were analyzed using ImageJ software. The relative reduction in fluorescence caused by heating was measured and presented as I/I0. Temperature change was calculated using the temperature coefficient obtained from the calibration.

### 2.11. Statistics

Experiments with cultured cells included at least three biological replicates. Data are presented as mean ± standard deviation, with the exception of image analysis results presented as mean ± standard error of the mean. A two-sample t-test was performed using ANOVA and Student’s t-test. In all statistical evaluations, *p* < 0.05 was considered statistically significant. For groups with unequal variances, a Mann–Whitney test was used. Origin 2022 (OriginLab Corporation, Northampton, MA, USA) software were used for statistical analysis.

## 3. Results

### 3.1. Preparation and Ultrastructural Analysis of the Hybrid Fibrous Materials

The cell scaffolds used in this work were composed of a fibrous nanomaterial attached to a glass plate, similar to our previous studies [43]. The nanomaterial consisted of aligned nylon nanofibers with an average diameter of 100 nm obtained by electrospinning method (Figure 2a). CuS-BSA TNPs were prepared by the biomineralization method proposed by Yang T. et al. [38]. The resulting particles had an average diameter of 22.7 ± 5.7 nm (Figure 2d). The UV-vis-NIR absorption spectrum of the CuS-BSA TNPs (Figure 2h) shows that the resulting particles effectively absorb in the NIR-I window. According to the authors of the original paper, particles of this size had a photoconversion efficiency of 51.5% when exposed to 808 nm irradiation.

An obligatory step in the preparation of the nylon scaffolds is plasma-chemical treatment, which simultaneously sterilizes the sample and promotes the adsorption of culture medium proteins, thus improving cell adhesion. This treatment induces surface oxidation accompanied, in the case of the linear polyamide nylon, by fragmentation of polymer chains with the formation of carboxyl groups. To improve the efficiency of TNPs immobilization, we developed a method that includes adsorption of PEI on plasma-oxidized nylon fibers. We based this on the assumption that a large number of electrostatic interactions between the chains of PEI and oxidized nylon would prevent desorption of the former from the surface. TNPs were immobilized on a PEI layer using glutaraldehyde as a homobifunctional cross-linker. To prevent protein desorption from the CuS surface, which would lead to degradation of TNPs, scaffolds were treated with an aqueous solution of sodium borohydride to reduce Schiff bases. Initial control of the immobilization efficiency was performed by AFM. The results of AFM and SEM image analysis of nylon-oriented fibers before and after TNPs grafting are shown in Figure 2.

In the control image obtained before the modification of the fibers, it can be seen that the surface of the fibers is smooth and homogenous. According to AFM data, surface modification resulted in efficient coating of the fibers with particles (Figure 2b). However, in this case a significant amount of adsorbed nanoparticles were also observed on the surface of the glass substrate. Obviously, a high background signal creates technical difficulties for assessing the temperature gradient created by TNPs on the fibers.

The second modification method made in this work involved inclusion of TNPs inside the fibers. For this, freshly prepared TNPs dispersed in formic acid were added to the nylon solution. The concentration of nanoparticles in the sprayed solution was 1.5% *w*/*v*, which corresponds to 12.5% mass fraction in the dry material of the fiber. Further increasing the TNPs concentration in the nylon solution reduced the durability of the fibers, which were torn in the process of drying on the collector, which disrupted the formation of fibrous material. As a result of the modification, fibers were effectively coated with TNPs, as shown in Figure 2c. The high resolution SEM image (Figure 2e) and elemental map reconstructions (Figure 2f,g) demonstrate the presence of particles incorporated inside the fibers onto AU-In scaffold. To demonstrate the retention of the size and shape of the nanoparticles, the material was treated for 60 s in plasma, which resulted in the removal of nylon. An AFM image of the particles left on the glass after plasma treatment is shown in the Appendix A. This method of modification reduces the number of stages of nanomaterial preparation, and also makes it possible to reduce background heating, since in this case there are no particles adsorbed on the glass substrate.

### 3.2. Photothermal Properties of Nanofibers

Micro-scale temperature measurement was performed by fluorescence thermal imaging. To evaluate the photothermal conversion efficiency of TNPs-modified scaffolds, we studied the dependence between Rhod B fluorescence intensity and the temperature on the fiber surface. Figure 3a illustrates linear increase in normalized fluorescence (I/I_0_) with temperature increase from 20 to 44 °C (−1.63%/°C).

We studied laser-induced heating of nanofibers under NIR irradiation (808 nm) at the power density of 3.0 W/cm^2^ for 5 min. As shown in Figure 3b, AU-In and AU-Coat scaffolds were heated by 10.5 ± 1.0 °C and 12.9 ± 5.0 °C, respectively (Figure 3b). The variability of the temperature values during irradiation of AU-Coat fibers is probably related to uneven coating of the fiber surface with nanoparticles and to the background heating caused by TNPs adsorbed on the glass plate. Non-modified AU fibers were heated to 7.6 ± 1.1 °C under the same conditions.

### 3.3. Effects of Photothermal Stimulation on Cell Viability of the SH-SY5Y Human Neuroblastome Cells

To assess the viability of differentiated SH-SY5Y cells on TNPs-decorated scaffolds under NIR irradiation, the cells were cultured for 24 h after photothermal exposure at a power density of 0.5–5.0 W/cm^2^, followed by LIVE/DEAD analysis (Figure 1A). Figure 4 does not show cell viability data in the absence of NIR stimulation since the presented TNPs-modified materials did not reduce cell viability of differentiated SH-SY5Y and were biocompatible. As shown in Figure 4a,b, NIR irradiation at 0.5 W/cm^2^ also did not reduce the viability of neuroblastoma cells on either of the scaffolds studied and the percentage of dead cells did not exceed 4%. At 1 W/cm^2^, the percentage of dead cells increased to 7% for AU-In scaffolds.

Significant cytotoxicity (up to 40%) of photothermal exposure was observed at 3 W/cm^2^ for cells cultured on AU-Coat scaffolds. However, the fraction of dead cells on AU-In scaffolds at the same power density still did not exceed 6–7%. Figure 4b does not show values for the power density 5 W/cm^2^, because at the given intensity of NIR irradiation the fraction of dead cells on all types of modified scaffolds exceeded 90%.

### 3.4. Heating Efficiency of Light-to-Heat Converting Scaffolds under NIR Irradiation and Measurement of Intracellular Temperature of Hippocampal Neurons

We analyzed laser-induced thermal doses delivered upon cell contact with photothermal scaffolds at a radiation intensity that does not lead to significant cell death. The intracellular temperature increment was quantified by epi-fluorescence microscopy imaging using Rhod B as a temperature-sensitive fluorescent dye. Prior to that, we measured the decrease in normalized fluorescence of Rhod B-loaded cells with temperature increase (−1.87%/°C, Figure 5a).

Figure 5b shows a representative time course of intracellular temperature increment. Without nanoheaters on the scaffold surface the temperature increase occurs more evenly over time, compared to TNPs-decorated nanofibers. Figure 5c demonstrates temperature distribution in the cells located on the surface of the scaffolds. NIR stimulation produces local heating of neurons on modified fibers. Figure 5d shows the maximum temperature change during irradiation for 5 min. Irradiation with the NIR laser under control conditions (cells on glass plate) leads to a dose-dependent increase in intracellular temperature, due to complete absorption of 808 nm radiation by water and glass. A significant increase in intracellular temperature was observed in rat hippocampal neurons interacting with the AU-In nanofibers (mean Δ*T* of 10.1 ± 3.2 °C at laser power density 1 W/cm^2^, and Δ*T* of 12.8 ± 3.6 °C at laser power density 3W/cm^2^). Neurons that were in contact with the AU-Coat surface under NIR irradiation, also showed a decrease in the fluorescence level corresponding to an increment in intracellular temperature (Δ*T* of 11.2 ± 3.3 °C at the laser power density of 1 W/cm^2^ and 9.9 ± 2.4 °C with 3 W/cm^2^).

We also measured the intracellular temperature of SH-SY5Y neuroblastoma cells. As shown in Figure 5e, the average change of intracellular temperature of SH-SY5Y cells cultured on the surface of particle-modified nanofibers was ~7 °C and ~14 °C for 0.5 W/cm^2^ and 1 W/cm^2^, respectively.

### 3.5. Effect of Photothermal Stimulation by Light-to-Heat Converting ECM-Mimetic Scaffolds on Neuronal Differentiation, Neurite Outgrowth and Elongation of Human Neuroblastoma Cell Line SH-SY5Y

We further assessed the ability of TNPs-modified scaffolds to stimulate neurite outgrowth and neuronal differentiation in human neuroblastoma cells mediated by fiber nanotopogy and heat-inducing properties. SH-SY5Y cells were seeded onto the surface of the fibrous scaffolds after 1 day of incubation in the Differentiation medium (see Methods). Then the cells were NIR-irradiated at the power density of 0.5 or 1 W/cm^2^ for 5 min and further incubated for 2 days (Figure 1B). Immunostaining was performed to detect and quantify differentiated neurons and to perform morphometric analysis.

Figure 6a shows that the cytoskeleton of the cells is oriented parallel to the nanofibers. It is especially noticeable for cells on AU scaffolds, in which the spatial distribution of actin stress fibers coincided with the orientation of the nanofibers. NIR stimulation did not affect the neurite orientation (which is probably more affected by surface topology). Thus, the representative Figure 6b graph shows that the peaks of angular distribution of neurites and of the nanofibers that they grow on coincide for all types of fibrous scaffolds. In the SH-SY5Y cells cultivated without NIR stimulation on modified and unmodified scaffolds, the proportion of β3-tubulin positive cells in the population was increased compared to the control plate (to 29.7 ± 8.2% for AU-In and to 11.4 ± 3.8 for control), as shown in Figure 6c. At the same time, NIR stimulation further enhances this trend: at the power density of 0.5 W/cm^2^, the proportion of differentiated cells on the control glass plate was doubled. For the cells cultivated on the AU-Coat substrate, it increased ~ 4-fold (from 17.8 ± 3.6% without NIR to 63.5 ± 22.1 and 43.9 ± 8.2% at 0.5 and 1 W/cm^2^ power density, respectively). A comparable effect was achieved when cells interacted with the AU-In scaffold under 1 W/cm^2^ irradiation. The proportion of differentiated cells increased twofold, from 29.7 ± 8.2% without NIR to 37.9 ± 8.1 and 63.2 ± 16.5% with 0.5 and 1 W/cm^2^ NIR, respectively. As seen in Figure 6d, fibrous scaffolds without NIR stimulation caused a slight increment in neurite length, with a mean neurite length of 43.0 ± 1.0 μm for unmodified fibers (AU), 42.6 ± 1.4 μm for AU-In and 40.5 ± 0.9 μm for AU-Coat. The mean neurite length of SH-SY5Y cells after irradiation increased to 60.6 ± 0.9 μm for AU-In and 48.5 ± 0.8 μm for AU-Coat at 0.5 W/cm^2^. Increasing the power density to 1 W/cm^2^ caused a reduction in the mean neurite length to 49.7 ± 1.1 μm with AU-In and an increase to 54.6 ± 1.1 μm with AU-Coat. It is noteworthy that non-modified AU fibers with NIR stimulation of 1 W/cm^2^ promoted differentiation as indicated by a statistically significant increase of the mean neurite length to 57.7 ± 1.4 μm. A more indicative assessment of the degree of neuron differentiation was performed by detailed analysis of neurite length distribution in the cell population. Figure 6e shows that the fraction of cells with long neurites increased from 13 and 14% (for 80 μm—the value of the bin center) to 22 and 16% and 9 and 6% (for 120 μm) for AU-In and AU-Coat, respectively, at 0.5 W/cm^2^. At the same time for the AU-In scaffold increasing power density to 1 W/cm^2^ there was a decrease in the proportion of neurites with a length of more than 40 μm (from 22% to 19% for 80 μm). For the AU-Coat scaffolds, more intensive NIR irradiation caused an increase in the proportion of neurites with a length of 120 μm or more, demonstrating the presence of cells with longer neurites indicative of highly differentiated neurons.

### 3.6. Effect of Photothermal Stimulation and Scaffolds Nanostructural Features on the Neurite elongation, Orientation and Branching of Hippocampal Neurons

We have previously shown that aligned ultrathin nanofibers are capable of accelerating the directional growth of neurites [43]. In the present work, we investigated the impact of nanofibers decorated with TNPs as nanometric topography cues and as local heating sources on neurite elongation and axonal branching of rat hippocampal neurons. After adhesion to the scaffold for 24 h, primary neurons were exposed to NIR (808 nm) irradiation at the specified power and then further cultivated for 6 days (Figure 1C).

Figure 7a shows immunefluorescent images of hippocampal neurons on various scaffolds with and without NIR stimulation.

Decorating the surface of nanofibers with nanoparticles did not affect the orientation of neurites which, like in the case of AU scaffolds, mainly remained oriented along the fibers. The maximal total elongation of neurites per cell without NIR stimulation was observed with AU-Coat scaffolds (605.3 ± 25.0 μm), which exceeded the control values (375.0 ± 27.0 μm) almost 2-fold (Figure 7b); however, NIR irradiation with this scaffold caused only a moderate increase in neurite length (676.0 ± 166.7 μm). Thus, it is primarily the additional nanoscale elements (nanoparticles) on the fiber surface that cause an increase in neurite outgrowth. In addition, the contact of neurons with the AU-Coat scaffolds leads to extensive branching and development of complex dendritic trees (Figure 7a). This effect was confirmed quantitatively by the analysis of the number of neurites per cell (Figure 7c). At the same time, the numbers of neurites per neuron during cultivation on either the control glass plate, non-modified AU or AU-In fibers were noticeably lower (Figure 7c). However, the effect of photothermal stimulation became dramatically more pronounced when neurons were cultured on the AU-In nanofibers. With this type of scaffold, elongation of neurites per cell increased 2-fold, from 314.2 ± 7.7 μm without stimulation to 625.0 ± 161.8 μm with NIR stimulation (Figure 7b). Figure 7d,e show that both modified (AU-In and AU-Coat) and non-modified fibers (AU) contributed to the overall cell survival in the neuroglial culture. Thus, while the fractions of neurons in the culture remained stable, the fraction of non-neuronal cells increased (Figure 7d). However, shown in Figure 7a, only AU-In type scaffold causes an increase in the astrocyte population upon irradiation. The fraction of apoptotic cells on the AU-In and AU-Coat scaffolds is significantly reduced with NIR stimulation (from 44.4 ± 10.1% and 43.3 ± 5.1% to 25.6 ± 9.4% and 6.7 ± 9.3%, respectively) compared to non-irradiated samples. The number of non-neural (proliferating) cells also increased. NIR exposure contributed to the survival and proliferation of astrocytes when cultured on a glass plate, which was demonstrated by an increase in the fraction of non-neuronal cells and a decrease in apoptotic cells in culture (Figure 7a,d,e).

## 4. Discussion

“Smart” materials, which combine the biological effects of surface nanoarchitecture with the ability to control cellular activity through local thermal exposure are very attractive not only for tissue engineering, but also for multiple other applications in translational medicine. In the current work, we present novel biomaterials composed of oriented nylon nanofibers with a diameter of 100 nm decorated with TNPs. CuS-BSA TNPs proposed as nanoheaters have significant absorption at 808 nm (Figure 2h), a wavelength which falls within the NIR-I window. The NIR light can penetrate deep into the tissues (>10 mm) and thereby offers a noninvasive tool to control specific cells remotely in vivo [48].

High biocompatibility, low biodegradation rate [40,49] and low cost of nylon-4,6, as well as its adhesive properties that eliminate the need for further modification with additional adhesive agents, make scaffolds of ultrathin nylon fibers highly applicable for in vivo use, and neural tissue engineering in particular. Previously, we have demonstrated superior efficiency of oriented ultrathin nylon nanofibers over submicron fibers for stimulating directed growth of neurites, as well as improved synaptogenesis and formation of connections between hippocampal neurons [43]. The electrospinning technique has a number of advantages for obtaining scaffolds consisting of polymer fibers: adjustable microarchitecture, large specific area of the fibrous material and the capacity for embedding nanoparticles, growth factors, etc. [26]. It is an effective approach for fabrication of fibrous materials of targeted design using a combination of electrospinning, electrospraying, and impregnation [7]. An important advantage of our setup for electrospinning is the capacity to produce free fibers that are not attached to the substrate. This allows the fibers to be transferred to any target substrates, regardless of their material, and also, potentially, to form bundles from individual fibers suitable for filling hollow nerve guidance conduits used for nerve regeneration. A similar design utilizing bundles of micron fibers inside a polymer tube has already been successfully applied for peripheral nerve regeneration in vivo [13]. Photothermal scaffolds made of fibrous materials were previously fabricated by electrospinning with the addition of light-converting particles to the polymer solution. The authors of these studies have also demonstrated the capacity of thermal exposure to efficiently destroy skin tumor cells, as well as to support cell adhesion, migration, and proliferation of normal skin cells in vitro. These qualities made it possible to successfully apply such nanomaterials to healing skin defects [50] and inhibiting the growth of skin tumors in vivo [26,51]. It should be noted, however, that the nanomaterials described in these works were composed of randomly oriented fibers with a diameter of 200–500 nm.

We applied two different methods for modifying fibers with nanoparticles. The first method involved fixation of TNPs on the surface of nylon nanofibers by chemical immobilization of CuS-BSA TNPs on the surface of nylon (AU-Coat fibers). However, part of the particles ended up being adsorbed on the glass substrate. Another feature of this method was gradual loss of particles from the scaffold surface during incubation in an aqueous solution (PBS) at 37 °C for a month. (Appendix A). These factors must be taken into account when choosing the irradiation mode for in vivo experiments. The second modification method involved incorporation of TNPs into fibers by adding nanoparticles to the polymer solution (AU-In fibers). The resulting nanofibers were affected by additional restrictions on the relative content of particles to polymer by mass. According to the study of Wang et al. [50], the reported efficiency of the electrospinning process used for production of fibers composed of poly(D,L-lactic acid)/poly(ε-caprolactone) was reduced when a mass fraction of added nanoparticles exceeded 50% wt. In our experiments, it was still possible to form fibers with the fraction of incorporated nanoparticles over 12.5% by mass and collect them on a rotating collector, but they were destroyed upon drying. This could be potentially explained by decreased mechanical durability of the fibers when a significant fraction of nanoparticles with a size comparable to the fiber diameter becomes embedded into the actual fiber. It has been noted that addition of nanoparticles to the polymer solution during electrospinning resulted in a significant decrease in the diameter of poly(L-lactic acid) and poly(D,L-lactic acid)/poly(ε-caprolactone) fibers [50,51]. Furthermore, addition of nanoparticles to a sprayed polymer solution can lead to the formation of beads on fibers, as observed with fibrous material produced from poly(3-hydroxybutyrate) [7]. However, the size of our fibers remained uniform and no beads were observed (Figure 2).

TNPs-modified scaffolds of types AU-In and AU-Coat demonstrated different photoconversion efficiency under NIR irradiation, which is associated with a different density of heating elements on the surface, depending on the decoration method (Figure 3). The surface temperature of the modified fibers exceeded the temperature of the unmodified fibers and the glass substrate (Figure 3). This confirms that the particles included in the scaffold retain their light-to-heat properties. As mentioned above, the modification process used for production of the AU-Coat samples left a pool of plasmonic particles adsorbed on the surface of the glass substrate besides the ones successfully immobilized on the fibers, which could further contribute to the observed temperature increase. In addition, it is difficult to achieve a uniform density of nanoparticles on the fiber surface with this method of modification. This resulted in variability of temperature values when irradiating AU-Coat fibers. These factors complicate the estimates and make it difficult to determine the relative efficiency of each photothermal nanomaterial type. The photothermal properties of scaffolds may also differ due to the fact that the heat generated by the particles located on the surface of the fibers and the incapsulated inside the fibers could be dissipated at different rates. The thermal conductivity of nanosized nylon fibers is much higher compared to bulk material [52]. So, considering high thermal conductivity of the fibers, the heat generated by the particles inside the fiber efficiently spreads through the fiber, which increases the dissipation rate due to the larger area of contact with air.

As we reported previously [42], nylon fibers can significantly increase the proliferation rate of neuroblastoma cells. Since even minor changes in the nanotopology of a material can cause significant changes in cell behavior [4], we tested the cytotoxicity of the nanoparticle decorated materials. We show that our novel photothermal nanomaterials are biocompatible and do not notably cause cell death in the absence of NIR stimulation (Figure 4). NIR stimulation with intensity ranging from 0.5 to 5 W/cm^2^ also did not lead to cell death on the control glass substrates. Significant cytotoxicity effects were observed upon irradiation at a power density of 5 W/cm^2^ for 5 min on all types of fibrous scaffolds. Similar cytotoxicity effects have been observed earlier at 5 W/cm^2^ with neural stem cells cultured on glass coverslips of modified gold nanocages [33]. At the same time, in the case of the AU-Coat type nanofibers, a significant cytotoxic effect is observed already at 3 W/cm^2^, which is consistent with the data on the maximum heating level achieved at a given radiation intensity according to Figure 3. Thus, high cytotoxicity of the modified scaffolds was apparently induced by overheating. Based on these observations, further study of photothermal modulation of cellular activity was carried out at irradiation intensities not exceeding 3 W/cm^2^.

Within the last few years, many groups reported development of photothermal conversion platforms for modulating neuronal activity [30,31]. Various nanomaterials are used as nanotransducers to convert light into heat: gold nanostructures, nanocrystalline semiconductors of transition metal sulfides, organic particles, carbon nanomaterials, etc. [53]. It should be noted that no temperature monitoring of cells in contact with light-stimulated materials was performed in these studies while assessing the cellular effects of photothermal stimulation. In [34], the authors measured cell temperature, but for photothermal stimulation they used intracellularly localized nanoheaters (polydopamine nanoparticles). However, with intracellularly delivered nanoheaters, it is impossible to accurately determine the thermal dose due to the inability to control the precise number of AuNPs within or in contact with the cells [33]. This points to a potential advantage of using light-to-heat scaffolds for controlled photothermal stimulation. In this work, we were the first to assess photo-induced intracellular heating of cells in contact with a plasmonic surface (Figure 5). The fibrous scaffolds modified with light-converting particles (AU-In and AU-Coat) are capable of locally heating the cells under NIR irradiation. The maximum values of intracellular temperature have been achieved on the AU-In scaffolds. Fibers containing nanoparticles inside might be able to provide more uniform heating, since the heating element is the entire surface of the fiber with high thermal conductivity. The observed variability in the change of intracellular temperature could be associated both with the heterogeneity of the fiber coating with nanoheaters (in the case of the AU-Coat scaffold) and with the varying functional state of the cells. There have been previous reports that thermal conductivity can vary significantly between live and dead cells, which, in turn, affects the ability to heat them [50]. Therefore, lower values of intracellular temperature achieved on the AU-Coat scaffolds may be associated with a significant cytotoxic effect, as shown in Figure 4. It is also known that the temperature of intracellular compartments varies significantly under normal conditions. Thus, the temperature of the cell nucleus is 1 °C higher than the temperature of the cytoplasm [54], while the temperature of mitochondria can be 10 °C higher [55]. A more prominent increase in the intracellular temperature of neuroblastoma cells exposed to irradiation of lower intensity may reflect greater thermal sensitivity of cancer cells compared to normal cells. The use of hyperthermia as an antitumor therapy is based on this feature of cancer cells [56]. The difference in the degree of heating between hippocampal neurons and neuroblasts can also be associated with a difference in thermal properties (thermal conductivity and thermal diffusivity) between healthy and tumor cells [57]. It has been shown that even among different cancer cell types there are significant differences in thermal properties [58].

It has been demonstrated earlier that incorporation of additional nanoscale topological stimuli into the composition of nanofiber scaffolds favorably affects the processes of neuronal differentiation and maturation [12,19]. We performed a detailed analysis of the nanotopogy-mediated and heat-induced capacities of TNPs-modified scaffolds to stimulate neurite outgrowth and neuronal differentiation in human neuroblastoma cells. In the absence of NIR stimulation, both modified and unmodified scaffolds with a fiber diameter of 100 nm caused an increase in the proportion of β3-tubulin positive cells in the population, but without a significant increase in the fraction of highly differentiated cells (Figure 6). We demonstrate for the first time that nanofibers with an average diameter of 100 nm can stimulate directed neurite outgrowth in differentiated SH-SY5Y cells. Comparing the organization of the cytoskeleton of cells cultured on ultrathin fibers or CB-NPs-modified nanofibers, it can be seen that cells on the AU scaffolds are characterized by the presence of a large number of actin stress fibers directed parallel to nanofibers. At the same time, cells on TNPs-modified scaffolds form higher numbers of filopodia and lamellipodia than ordered stress fibers. Presumably, this may be caused by increased nano roughness of the scaffold surface due to nanoparticles. Overall, it allows us to conclude that nanofiber scaffolds with an average fiber diameter of 100 nm orchestrate cytoskeletal reorganization by templating surface topography.

The stimulation effect of nanotopology on neurite elongation has been studied in PC-12 cells cultivated on polyurethane fibers with the diameter of 519 ± 56 nm coated with the 50 nm gold nanoparticles [19] and 260 ± 70 nm polycaprolactone-gelatin fibers coated with the 10 nm gold nanoparticles of [12]. However, the nanomaterials used in the works cited above had a significantly larger fiber size than those described in the present study. Thus, our work shows for the first time that contact guidance fibers, 100 nm in diameter decorated with CB nanoparticles of about 22.7 ± 5.7 nm can provide directed growth of differentiated cell processes. Since NIR stimulation did not affect the orientation of the processes, it is plausible that cell alignment is determined by the scaffold’s nanotopology.

Previously, photo-induction of neurite growth by NIR radiation has been shown using intracellularly localized heat sources. Thus, in the article by Paviolo et al., cells of the NG108-15 neuronal line with internalized gold nanorods developed a significantly higher percentage of neurons with neurites upon exposure to laser irradiation (780 nm) [59]. The use of extracellularly localized nanotransducers on a 2D substrate for photothermally induced growth was first described by Akhavan et al. [60]. It was shown that culturing human neural stem cells on reduced graphene oxide nanomeshes with NIR stimulation resulted in intense cell differentiation, with more pronounced cell elongation and preferential differentiation into neuronal rather than glial lineage. Similarly, in the context of neuronal differentiation, Jung S. et al. showed that laser-induced thermal stimulation using glass coverslips coated with gold nanocages enhanced neuronal differentiation of rat neural stem cells [33].

This work is the next step in the development of light-to-heat converting materials for neuromodulation. We have obtained 3D fibrous materials with light-to-heat converting properties that enhance neurite outgrowth and promote neuronal maturation under NIR irradiation (Figure 6). Our results show that ultrastructure of fibrous materials increases the number of neurons during differentiation, and photothermal stimulation promotes neurite elongation and increases the percentage of highly differentiated cells.

We have studied the effect of nanoparticles, as nanometric topography cues on the surface of ECM-mimetic scaffolds, and as local heat sources on neurite elongation and axonal branching of rat hippocampal neurons (Figure 7). It should be noted that the modified scaffolds do not appear to have a toxic effect on cultured primary neuroglial cells, with or without NIR irradiation at the chosen intensities. At the same time, while the fraction of neurons in culture remained stable, the fraction of non-neuronal cells increased. Shah et al. previously demonstrated that mechanical stimulation of neural stem cells using graphene-nanofiber hybrid scaffold induced their selective differentiation into mature oligodendrocytes in the absence of differentiation inducers in the culture media [60]. It could be suggested in our study that nanoscale surface cues might specifically favor the survival of non-neural cells. However, the nature of this phenomenon requires further investigation.

Similar to the results with differentiated neuroblastoma cells, the orientation of neurites in primary neurons was also unaffected by decoration of the surface of nanofibers with TNPs in. At the same time, the contact of neurons with the surface of the AU-Coat substrate led to intense dendritic arborization and development of complex dendritic trees, which was reflected by an increase in the number of neurites per cell (Figure 7c). This could be due to the presence of a large number of nanoparticles on the surface of the AU-Coat scaffold and glass plate, which acted as additional topography cues and thus enhanced neurite branching. Similar to the results presented by Jung et al. [33] there was no significant change in the number of neurites in response to laser irradiation when hippocampal neurons were cultured on the AU-Coat scaffold. The interaction of cells with these substrates initiated the maximal observed neurite elongation regardless of NIR stimulation. Thus, coating the nanofiber surface with additional nanosized elements (nanoparticles) can enhance neurite outgrowth. The effect of photothermal stimulation was particularly pronounced when neurons were cultured on impregnated TNPs nanofibers; in this case NIR irradiation doubled the length of neurites.

The case of creating plasmonic ECM-mimetic materials has been repeatedly presented by researchers in studies devoted to light/heat control of neuronal activity [33,61]. Researchers were also trying to elucidate whether the heat generated by activated surface plasmon structures actually had a stimulating/inhibiting effect on neurons or whether it was due to other factors, such as direct action of light, similar to photobiomodulation based by low-level laser therapy, or surface properties. In this work, we tried to separate the effects of NIR stimulation and topology-mediated effects on cell behavior. In addition, we measured the change in intracellular temperature during NIR irradiation and recorded an increase in temperature during the activation of plasmonic nanoparticles under exposure to light and the absence of a significant temperature increase in their absence in control samples and on unmodified nanofibers. Temperature data correlate with biological effects observed in cells (growth of processes and differentiation), suggesting that these effects are associated with photo-induced heating.

Several studies reported preparation and successful application of artificial nerve conduits based on electrospun fibrous materials in biological test systems [13,62,63]. However, relatively few works are deploying fiber nanomaterials with fiber diameters of 100 nm due to technical difficulties of manufacturing [42]. The use of composite light-converting nanomaterials demonstrates efficiency in stimulating differentiation of neural cells [59,60]; therefore, we believe that the combination of topology-mediated stimulation with thermally induced control of cellular activity is of practical interest for the creation of neurosurgical implants. Funnell et al. [64] used the combination of magnetic fields, magnetic nanoparticles, and aligned electrospun fibers to enhance neurite outgrowth. The authors demonstrated that combining the alternating field with magnetic nanoparticle-grafted fibers does not affect neurite outgrowth compared to control but improves outgrowth compared to freely dispersed magnetic nanoparticles; however, the size of the described fibers (2 µm) greatly exceeded the dimensions of the characteristic structural components of the neural tissue ECM. The group of Ishiwata S. [65] showed accelerated neurite outgrowth in the field of a local temperature gradient; however, the use of an optical heater (heating water with a laser with a wavelength of 1455 nm) is not applicable in vivo. Several studies [33] have shown the stimulating effect of heat on growth of neuronal processes using plasmonic materials, but the presence of nanoparticles as the only kind of ECM-mimetic topology stimuli did not allow to completely recreate the fiber ECM microenvironment of nerve tissue. In this work, we implemented both approaches: ECM-mimetic nanofibers, and local heating using NIR irradiation, suitable for use in vivo due to sufficient deep tissue penetration. The results of our in vitro studies show promise for further experiments focused on production of conduits based on composite light-converting materials for in vivo application. The biomaterials presented in our work possess unique features affecting the behavior and growth of nerve cells, which need to be taken into account in further studies. The decoration method based on the inclusion of thermoplasmonic particles inside the fiber seems to be more promising for manufacturing of nerve conduits. The advantages of this method include a reduction in the number of stages necessary for the manufacture of the conduit: nanoparticle decoration occurs simultaneously with the formation of fibers, and the resulting fibers are not fixed on the substrate, which simplifies the formation of yarn for filling the artificial nerve conduit.

## 5. Conclusions

A composite light-transforming material was created using 100 nm oriented fibers, decorated with plasmonic nanoparticles, and its ability to heat was studied. It is shown that despite the limitations on the load of thermoplasmonic particles, even the available amount is sufficient to create additional topological and thermal factors stimulating the directed growth of neurites. The development of scaffolds with optimal topology and incorporated nanoheaters, which additionally induce directed thermal action, will make it possible to obtain new stimuli-responsive materials for tissue engineering. The hybrid nanomaterials presented in this study have the potential for both “passive” (topographic) and remote “active” (thermal) stimulation. We performed a comparative analysis of photothermal scaffolds obtained by two methods—impregnation of light-to-heat converting nanoparticles inside fibers (AU-In) and surface coating with them (AU-Coat). It has been shown that AU-In scaffolds provide more uniform heating, which is manifested in the maximum observed intracellular temperature during photothermal treatment. In addition, these substrates make it possible to use a more intense NIR exposure (3 W/cm^2^) in the absence of a cytotoxic effect, in contrast to AU-Coat, which exhibit a cytotoxic effect at a given power density, which is possibly due to additional heating of the particles adsorbed on the surface of the glass substrate. Neuron cultivation on AU-In scaffolds significantly stimulates neurite outgrowth and increases the percentage of highly differentiated cells of human neuroblastoma at lower (0.5 W/cm^2^) power density of NIR irradiation than AU-Coat substrates. However, the interaction of rat hippocampal neurons with the AU-Coat substrate results in strong branching, an increase in the number of neurites per cell, and significant neurite elongation independent of NIR stimulation, while in the case of AU-In a similar effect can be achieved only with photo-induced heating of the cells.

The effects induced by our novel scaffolds in various cell types, namely enhanced differentiation of neuronal precursors and axon elongation, with and without photothermal stimulation, provide a basis for further development of perspective materials with combined topology- and thermal-induced stimulation properties for the development of artificial neural conduits for reconstructive neurosurgery.

## Figures and Tables

**Figure 1 nanomaterials-12-02166-f001:**
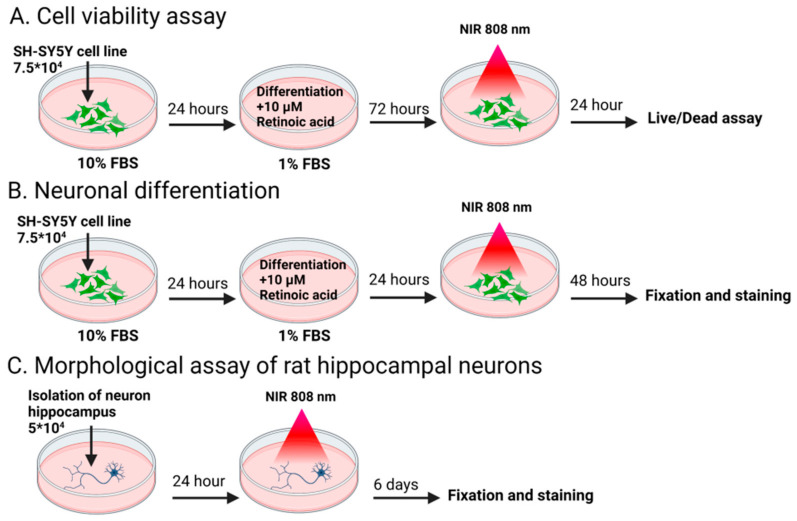
Experimental procedure for cell manipulation: scheme illustrates procedure viability assay (**A**), neuronal differentiation of human neuroblastoma SH-SY5Y cells (**B**), analysis of neurite elongation of rat hippocampal neuron (**C**) under NIR stimulation.

**Figure 2 nanomaterials-12-02166-f002:**
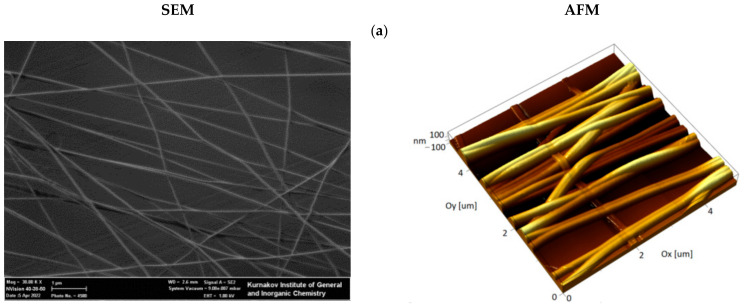
SEM and AFM images of AU nylon nanofibers: (**a**) non-modified (AU) fibers; (**b**) fibers coated with immobilized TNPs (AU-Coat); (**c**) fibers with incorporated nanoparticles TNPs (AU-In). Scale bar –1 μm. SEM image of the CuS-BSA TNPs (**d**). SEM/EDS image (**e**) and the elemental map reconstructions (**f**,**g**) of AU-In scaffold. UV-vis-NIR absorption spectrum of the CB NPs (**h**). First NIR window marked in green.

**Figure 3 nanomaterials-12-02166-f003:**
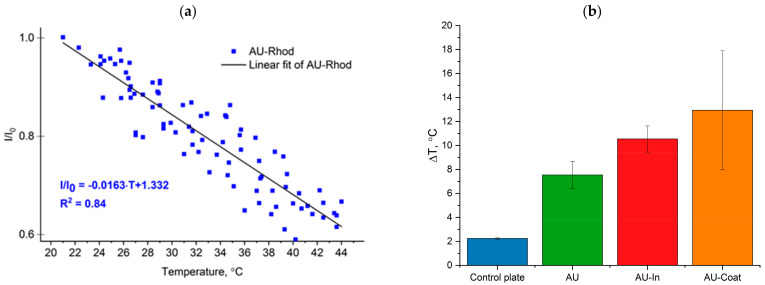
Photothermal properties of fibrous scaffolds. Temperature-dependent changes of relative fluorescence intensity (I/I0, I0 is fluorescent intensity at 20 °C) of Rhod B on the nylon AU fibers surface (**a**). Temperature changes at the surface of scaffolds under NIR irradiation (808 nm) at power density of 3.0 W/cm^2^ for 5 min (**b**).

**Figure 4 nanomaterials-12-02166-f004:**
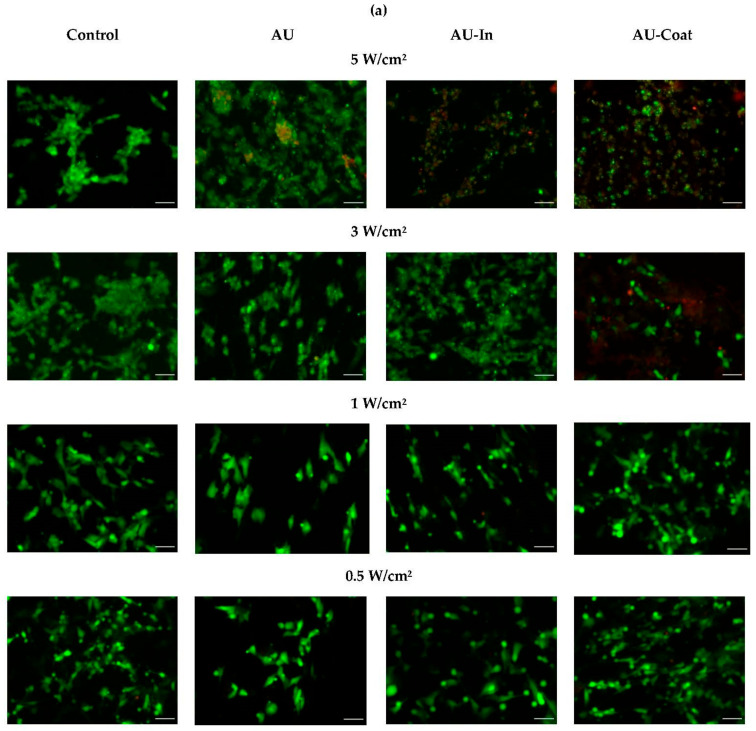
Cell viability of differentiated SH-SY5Y cells grown on AU and thermo-plasmonic nanofibers and exposed to NIR irradiation at different power density (0.5, 1, 3, and 5 W/cm^2^ for 5 min). Representative fluorescence images of the LIVE/DEAD analysis for 24 h after NIR exposition (**a**). Green—live cells stained with calcein AM, red—dead cells stained with propidium iodide. Scale bar—50 μm. Bars show relative fractions of live and dead cells as an average of three independent experiments (**b**).

**Figure 5 nanomaterials-12-02166-f005:**
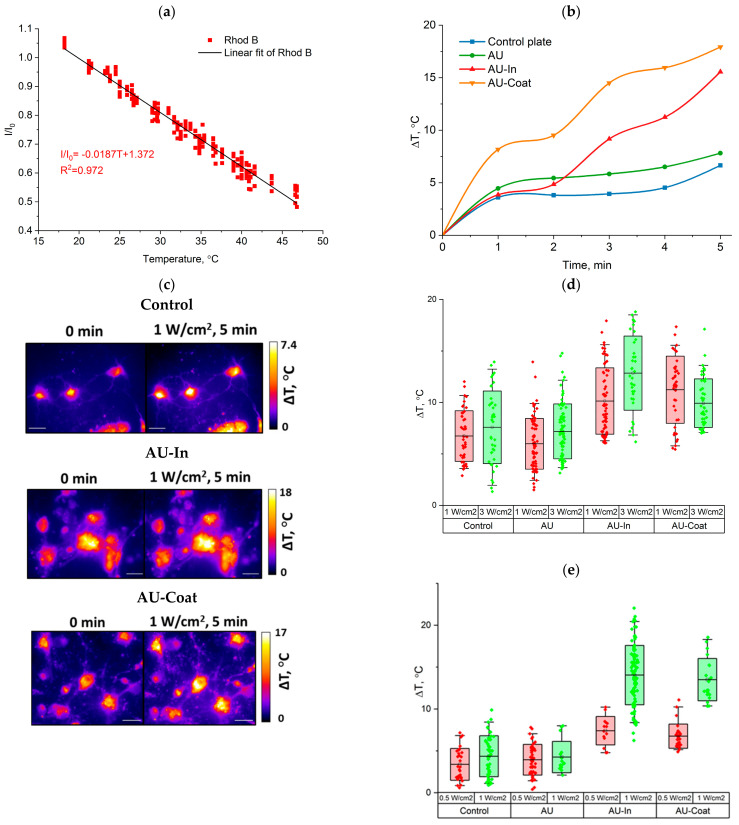
Analysis of intracellular temperature dynamics in rat hippocampal neuron during NIR stimulation. Temperature-dependent changes of normalized fluorescence intensity (I/I0, I0 is fluorescent intensity at 20 °C) of Rhod B inside the cells (**a**). Representative time courses of ΔT (°C) during NIR stimulation (**b**). Representative thermal changes in neurons before (0 min) and after 5 min of NIR stimulation. ΔT is represented in pseudo-colors. Scale bar—20 μm. (**c**). Temperature changes (ΔT) during NIR stimulation (1 and 3.0 W/cm^2^ for 5 min) in rat hippocampal neurons (**d**) and neuroblastoma cells (**e**), growing on different scaffolds.

**Figure 6 nanomaterials-12-02166-f006:**
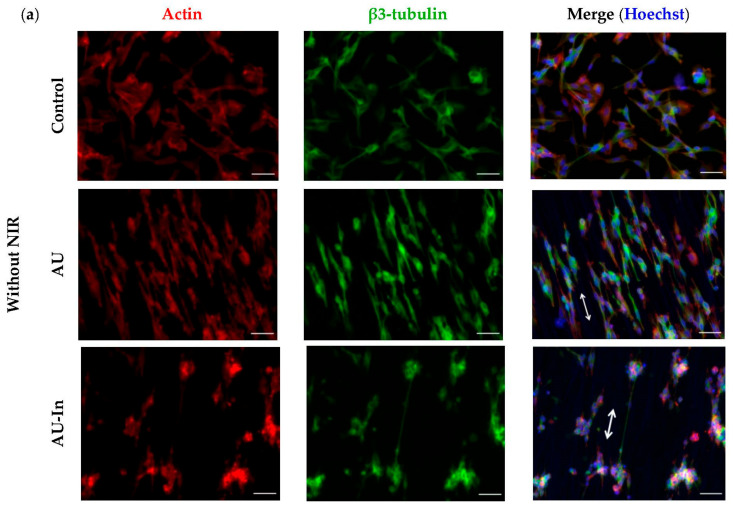
Neuronal differentiation of SH-SY5Y neuroblastoma cells grown on different scaffolds with/without NIR stimulation. Representative epifluorescence images of the SH-SY5Y cells 2 days after NIR stimulation (0.5 W/cm^2^ for 5 min) stained for β3-tubulin in green, phalloidin (as an actin network marker) in red, and nuclei in blue (**a**). Arrows show the direction of fiber orientation. Scale bar—50 μm. Representative angular distribution of fibers and cell neurites on scaffolds after NIR irradiation (808 nm) at 0.5 W/cm^2^ for 5 min (**b**). Comparison of the percentages of differentiated β3-tubulin-positive SH-SY5Y cells (**c**; values represent the means ± SD; * statistically significant difference, *p* < 0.05), mean neurite length (**d**; values represent the means ± SE) and frequency distribution of the neurite length (**e**), (each color represents the percentage of neurite frequency given length per bin).

**Figure 7 nanomaterials-12-02166-f007:**
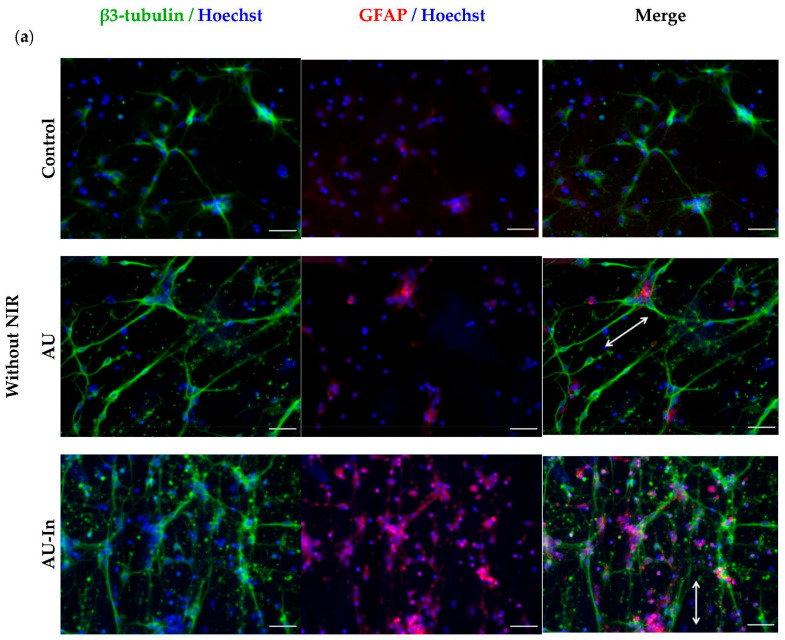
Nanotopology-mediated and photothermal stimulation of neurite outgrowth. Immunofluorescence images of neurons on different scaffolds with and without NIR irradiation stained for β3-tubulin in green, glial fibrillar acidic protein (GFAP) in red and nuclei in blue. The arrows show the direction of fiber orientation. Scale bar—50 μm (**a**). Comparison of the total elongation of neurites per cell (**b**), number of neurites per neuron (**c**), fraction of β3-tubulin-positive (neuron) cells (**d**), and percentage of live cells per total cell population (**e**).

## Data Availability

The data presented in this study are available in Appendix A.

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
