# Peer review of "Light-to-Heat Converting ECM-Mimetic Nanofiber Scaffolds for Neuronal Differentiation and Neurite Outgrowth Guidance"

_nanomaterials, 2022, doi:10.3390/nano12132166_

Round 1
Reviewer 1 Report
Authors did fantastic work using light for the enhancement of neuronal growth and guidance. I suggest to resolved following issues before publication.
1. authors chose the non-degradable nylon 6 fiber for the study. My concern is that is it really applicable in in-vivo conditions?
2. what sizer glass plates were used for culture?
3. How did culture do in nanofibers? did the author attach nanofiber to well ? Please explain in material method section in detail
4. what temperature was induced with light irradiated at different power densities ? what temperature was used for the stimulation of cells growth?
5. Are the temperatures more than 37 degrees compatible with the cells internal structure?
6. Authors are suggested to see reports related to near infrared light triggered photothermal effect and axon guidance on nanofibers.
ACS Applied Bio Materials 4 (12), 8424-8432, ACS applied materials & interfaces 10 (24), 20256-20270, Scientific reports 9 (1), 1-13
Author Response
Point 1. Authors chose the non-degradable nylon 6 fiber for the study. My concern is that is it really applicable in in-vivo conditions?
Response 1. Appropriate explanations have been introduced to the discussion. High biocompatibility, low biodegradation rate [Naoko Yamano, Norioki Kawasaki, Sayuri Ida, Yasuhide Nakayama, Atsuyoshi Nakayama, Biodegradation of polyamide 4 in vivo, Polymer Degradation and Stability, V. 137, 2017, pp 281-288, https://doi .org/10.1016/j.polymdegradstab.2017.02.004] and low cost of nylon-4,6, as well as its adhesive properties which eliminate the need for further modification with additional adhesive agents, make scaffolds of ultrathin nylon fibers highly applicable for in vivo use, neural tissue engineering in particular. In our earlier work [Antonova O.Y., Kochetkova O.Y., Shlyapnikov Y.M. ECM-Mimetic Nylon Nanofiber Scaffolds for Neurite Growth Guidance // Nanomaterials. 2021; 11(2):516] we discussed the biocompatibility and biodegradability of nylon 4/6.
Point 2. What sizer glass plates were used for culture?
Response 2. The plates had a size of 9x11x1 mm. The size was added to the method section.
Point 3. How did culture do in nanofibers? did the author attach nanofiber to well ? Please explain in material method section in detail
Response 3. Prepared electrospun fibers were transferred to sterile glass plates with a size of 9x11x1 mm and then glued at the edges with a polystyrene solution in toluene. Extended scaffold fabrication description has been introduced to methods. Photo of a composite scaffold has been added to Supplementary Materials.
Point 4. What temperature was induced with light irradiated at different power densities ? what temperature was used for the stimulation of cells growth?
Response 4. When choosing an irradiation mode, we had to find a fine line between stimulating action and heating leading to cell death (as is more often used in photothermal ablation of tumor cells). When measuring intracellular temperature, we detected changes in rhodamine fluorescence in different ROIs inside the cell. It is known that the temperature of intracellular compartments varies significantly under normal conditions. Thus, the temperature of the cell nucleus is higher than the temperature of the cytoplasm [Okabe, K., Sakaguchi, R., Shi, B. et al. Intracellular thermometry with fluorescent sensors for thermal biology. Pflugers Arch - Eur J Physiol 470, 717–731 (2018). https://doi.org/10.1007/s00424-018-2113-4], and the temperature of mitochondria can be 10°C higher [Chrétien D, Bénit P, Ha H-H, Keipert S, El-Khoury R, Chang Y-T, et al. (2018) Mitochondria are physiologically maintained at close to 50 °C. PLoS Biol 16(1): e2003992. https://doi.org/10.1371/journal.pbio.2003992]. In addition, the temperature of cells during NIR stimulation can significantly depend on the functional state of the cells. The variability of cell heating can also depend on the density (number) of heated elements in contact with its surface, in this case, fibers with impregnated particles and with particles on the surface of the fibers. We measured the intracellular temperature upon irradiation of hippocampal neurons with power densities of 1 and 3 W/cm2, which resulted in a significant temperature change compared to the control. Using a higher power density (5 W/cm2) as we have shown in the viability study results in cell death. On fig. 5d shows data on the increase in intracellular temperature during NIR stimulation of the specified power density for 5 minutes. Thus, a photoinduced temperature increase of about 10°C was used to stimulate hippocampal neurons. We added to Figure 5e data on measuring the intracellular temperature of neuroblastoma cells during NIR stimulation of 0.5 W/cm2 and 1 W/cm2 used to study thermo-induced differentiation. As can be seen, the change in the average intracellular temperature of SH-SY5Y cells cultured on the surface of particle-modified nanofibers was about 7 and 14°C for 0.5 W/cm2 and 1 W/cm2, respectively. Appropriate explanations have been added to results and discussion sections (p.11-12, 21).
Point 5. Are the temperatures more than 37 degrees compatible with the cells internal structure?
Response 5.The effect of local short-term heating on the internal structure was not investigated in the present work. In the future, we plan to investigate the effect of local heating on such structures, in particular, on the cytoskeleton and focal adhesions. In study [Polydopamine Nanoparticles as an Organic and Biodegradable Multitasking Tool for Neuroprotection and Remote Neuronal Stimulation Matteo Battaglini, Attilio Marino, Alessio Carmignani, Christos Tapeinos, Valentina Cauda, Andrea Ancona, Nadia Garino, Veronica Vighetto, Gabriele La Rosa, Edoardo Sinibaldi, and Gianni Ciofani ACS Applied Materials & Interfaces 2020 12 (32), 35782-35798 DOI: 10.1021/acsami.0c05497] the authors detected an increase of intracellular temperature during NIR stimulation using intracellularly localized polydopamine nanoparticles as nanoheaters and fixed Ca2+ influx at temperature values similar to ours, which indicates the preservation of the functional activity of cells. In present work we used a short-term exposure of cells to NIR radiation (5 minutes), which is significantly less than, for example, the photothermal treatment time used to achieve ablation of tumor cells - 30-60 minutes [Hannon, G., Tansi, F.L., Hilger, I and Prina-Mello, A. (2021), The Effects of Localized Heat on the Hallmarks of Cancer. Adv. Therap., 4: 2000267. https://doi.org/10.1002/adtp.202000267].
Point 6. Authors are suggested to see reports related to near infrared light triggered photothermal effect and axon guidance on nanofibers. ACS Applied Bio Materials 4 (12), 8424-8432, ACS applied materials & interfaces 10 (24), 20256-20270, Scientific reports 9 (1), 1-13
Response 6.We have introduced into the manuscript information about suggested reports (ref. 14, 18).

Reviewer 2 Report
In this article, authors introduce the photothermal effect to the nanofiber scaffolds that have been previously published for guided neurite growth in tissue engineering. Authors suggest two methods for implementing the photothermal effect by either surface coating the nanofiber with a kind of plasmonic nanoparticles or inserted into the nanofiber when the fiber material is prepared. Authors made a lot of efforts precisely characterizing the local, microscale temperature changes by the photothermal effect using the fluorescence imaging based technique. With not very common usage of the photothermal effect in neuronal cell engineering and studies and suggested photothermal tissue engineering platforms, it is interesting to see how these new methods can be used beyond this study in the near future. In addition, a lot of other works in photothermal effect do not show or try the characterization of microscale temperature changes, which is indeed much more relevant to the cell biology than large scale averaged temperature measurement data (e.g. thermographic camera). Despite the novelty and extensive experimental data, the reviewer thinks this manuscript can improve further with some revisions.
Here are some comments and questions to the authors:
· Why is it called LCNP? Light converting nanoparticle? There was no full name for the LCNP despite the repeated usage. Why not using photothermal NP or thermoplasmonic NP, the terms more commonly used instead? Is there any fundamental difference from those terms?
· What is the absorption spectra of the nanoparticles used in this work? Are these materials supposed to efficiently interact with NIR specifically?
· Related to the previous question, why is the nanoparticle photothermal effect very small (Fig. 3b)? Yes, the AU1 & AU2 conditions showed higher delta T, but compared with the non nanoparticle sample, additional thermal effect by the incorporated nanoparticles is subtle. The reviewer wonders if it is because the nanoparticles used are not designed for NIR absorption, nor have too low density. Can authors suggest how to improve further to reduce the laser power for the same consequences. Though it is NIR, compared to optogenetics etc lower laser power would be more attractive.
· Despite the effort to characterize the temperature, the reviewer wonders if the measured temperature changes in Fig. 3 is supposed to be identical to the changes in Fig. 4 in the same laser power density? In other words, do we have to conclude that average temperature change 12.5 degC of AU2 in Fig. 3 is the reason and the value is critical for the lower cell viability given in Fig. 4?
From a previous experience, without exactly the same environmental condition (e.g., substrate thickness, volume of culture medium etc.), the temperature change varies due to the change of different thermal equilibrium. Can authors comment on that?
· Naming: Authors named the suggested two methods as AU1 and AU2. However, the reviewer found it very difficult to follow which is which and confusing throughout the manuscript. The reviewer strongly suggests using an acronym which is more intuitive (for example, AU-Coat, AU-In). More importantly, it is not clear if the naming was consistently used in the manuscript. There’s a concern that it was defined differently as follows: AU1 was a surface coated NP sample in the discussion, but on Line 164, AU1 was the NP incorporated sample… Which is correct?
o Discussion: fixation of LCNPs on the surface of nylon nanofibers by chemical immobilization of CB NPs on the surface of nylon (AU1 fibers).
o Line 164, Aligned ultrathin nylon fibers with incorporated nanoparticles (Type AU1) were prepared by adding freshly prepared CB-NPs (1.5 % w/v) to nylon solution and over-night rotational mixing.
o Fibers coated with nanoparticles (Type AU2) were prepared by chemical immobilization of freshly prepared CB-NPs onto AU fibers glued to 168 glass plate.
· Lastly, while the new platform from an engineering point of view is novel and interesting, the implication of the biological experiments is not clear if it is significantly important or if the results deliver new observation or theory. If not mistaken, only the Fig. 7b shows some significant difference in the case of AU1 on whether the photothermal effect is significantly meaningful or not.
Here are also some minor but also important comments/suggestions:
· On line 236, 424, 425, there are some non-English words and characters. There seem to be more throughout the manuscript, which needs correction and consistency.
· On line 424, reference was not completed properly.
Author Response
Point 1. Why is it called LCNP? Light converting nanoparticle? There was no full name for the LCNP despite the repeated usage. Why not using photothermal NP or thermoplasmonic NP, the terms more commonly used instead? Is there any fundamental difference from those terms?
Response 1. We agree with the Reviewer and have changed the abbreviation to the proposed "thermoplasmonic nanoparticles" (TNPs). Explanation of the abbreviation was added to the introduction text.
Point 2. What is the absorption spectra of the nanoparticles used in this work? Are these materials supposed to efficiently interact with NIR specifically?
Response 2. We have introduced the UV-vis-NIR absorption spectrum of the CuS-BSA nanoparticles to Fig. 2h. As can be seen, the resulting particles have sufficient absorption at 808 nm. This wavelength is included in the NIR-I window, at which the absorption by water is minimal and the radiation openetrates deep enough into tissues [Chen, G., Cao, Y., Tang, Y., Yang, X., Liu, Y., Huang, D., Zhang, Y., Li, C., Wang, Q., Advanced Near-Infrared Light for Monitoring and Modulating the Spatiotemporal Dynamics of Cell Functions in Living Systems. Adv. Sci. 2020, 7, 1903783. https://doi.org/10.1002/advs.201903783]. Description of the spectrum was added to Materials and Methods, Results and Discussion (p. 4, 7, 8, 20).
Point 3. Related to the previous question, why is the nanoparticle photothermal effect very small (Fig. 3b)? Yes, the AU1 & AU2 conditions showed higher delta T, but compared with the non nanoparticle sample, additional thermal effect by the incorporated nanoparticles is subtle. The reviewer wonders if it is because the nanoparticles used are not designed for NIR absorption, nor have too low density. Can authors suggest how to improve further to reduce the laser power for the same consequences. Though it is NIR, compared to optogenetics etc lower laser power would be more attractive.
Response 3. Absorption spectrum of the thermoplasmonic nanoparticles (introduced to Fig. 2h) confirms the significant absorption at a wavelength of 808 nm. In addition, NIR stimulation of cells and measurement of intracellular temperature in the presence of nanoheaters on the fiber surface also confirm the thermal effect (Fig.5d and f). According to a study of the influence of irradiation regimes on cell viability (Fig. 4), an increase in heating (increase in power density) leads to cell death, which is comparable to the work [Jung, S.; Harris, N.; Niyonshuti, I.I.; Jenkins, S.V.; Hayar, A.M.; Watanabe, F.; Jamshidi-Parsian, A.; Chen, J.; Borrelli, M.J.; Griffin, R.J. Photothermal Response Induced by Nanocage-Coated Artificial Extracellular Matrix Promotes Neural Stem Cell Differentiation. Nanomaterials 2021, 11(5), 1216. doi: 10.3390/nano11051216.]. Increasing the number of nanoparticles in the composition of the fiber could help reduce the power to achieve the desired heating, however, an increase in the fraction of nanoparticles leads to a deterioration in the mechanical strength of the fibers due to their small diameter. Preservation of the fiber diameter in this case is a priority, because their size is mimetic ECM of nerve tissue. Nevertheless, the results obtained on the differentiation of neuroblastoma and the growth of processes of hippocampal neurons demonstrate a biological photothermal effect. In addition, it is possible to use particles with more efficient photoconversion, for example, gold nanostructure, but these nanoheaters can be more expensive, less biodegradable, and have low photostability, which is critical for the possibility of implementing cyclic (repeated) exposure.
Point 4. Despite the effort to characterize the temperature, the reviewer wonders if the measured temperature changes in Fig. 3 is supposed to be identical to the changes in Fig. 4 in the same laser power density? In other words, do we have to conclude that average temperature change 12.5 deg C of AU2 in Fig. 3 is the reason and the value is critical for the lower cell viability given in Fig. 4? From a previous experience, without exactly the same environmental condition (e.g., substrate thickness, volume of culture medium etc.), the temperature change varies due to the change of different thermal equilibrium. Can authors comment on that?
Response 4. Changes in scaffold temperature during NIR irradiation in air are shown in Fig. 3 are given to demonstrate that the material has photothermal properties. Measurement of intracellular temperature during NIR exposure and analysis of the effect of NIR radiation on cell viability/ morphology were performed in culture medium and under other conditions that were strictly controlled, since culture medium volume, beam diameter, and power density affect the efficiency of intracellular heating. To determine the critical temperature, we can use Fig. 5d and f where the change in intracellular temperature for different types of cells is presented when irradiated with the indicated power for 5 min. Based on these data, it can be assumed that the critical increase in the average intracellular temperature is more than 13°C. We did not measure the intracellular temperature during irradiation with higher power density values, because at 5 W/cm2 high cell death was observed. In study [Jung, S.; Harris, N.; Niyonshuti, I.I.; Jenkins, S.V.; Hayar, A.M.; Watanabe, F.; Jamshidi-Parsian, A.; Chen, J.; Borrelli, M.J.; Griffin, R.J. Photothermal Response Induced by Nanocage-Coated Artificial Extracellular Matrix Promotes Neural Stem Cell Differentiation. Nanomaterials 2021, 11(5), 1216. doi: 10.3390/nano11051216], authors detected an increase in the temperature of the culture medium by 35°C at a power density of 5 W/cm2 on gold nanocages attaching to glass coverslips which resulted in a significant lethal effect on rat fetal neural stem cells. However, it is impossible to directly compare these data with ours, since culture medium temperature and intracellular temperature may differ.
Point 5. Naming: Authors named the suggested two methods as AU1 and AU2. However, the reviewer found it very difficult to follow which is which and confusing throughout the manuscript. The reviewer strongly suggests using an acronym which is more intuitive (for example, AU-Coat, AU-In). More importantly, it is not clear if the naming was consistently used in the manuscript. There’s a concern that it was defined differently as follows: AU1 was a surface coated NP sample in the discussion, but on Line 164, AU1 was the NP incorporated sample… Which is correct?
Discussion: fixation of LCNPs on the surface of nylon nanofibers by chemical immobilization of CB NPs on the surface of nylon (AU1 fibers).
Line 164, Aligned ultrathin nylon fibers with incorporated nanoparticles (Type AU1) were prepared by adding freshly prepared CB-NPs (1.5 % w/v) to nylon solution and over-night rotational mixing.
Fibers coated with nanoparticles (Type AU2) were prepared by chemical immobilization of freshly prepared CB-NPs onto AU fibers glued to 168 glass plate.
Response 5. We agree with the Reviewer and have changed the abbreviations to the proposed AU-Coat for surface coated scaffolds and AU-In for incorporated ones. Explanations of the abbreviation were added to the introduction text. Corrections have been introduced to the marked lines.
Point 6. Lastly, while the new platform from an engineering point of view is novel and interesting, the implication of the biological experiments is not clear if it is significantly important or if the results deliver new observation or theory. If not mistaken, only the Fig. 7b shows some significant difference in the case of AU1 on whether the photothermal effect is significantly meaningful or not.
Response 6. The results obtained in this work show that the effect of thermal stimulation is manifested not only in the elongation of the axons of hippocampal neurons, as can be seen in Fig. 7b, but also in the increase in the average length of the processes, the b3-tubulin-positive fraction and highly differentiated neuroblastoma cells, as seen in Fig. 6c, d and e. The insignificant stimulatory effect of NIR iraadiation and local heating upon contact of neurons with AU-Coat scaffolds is apparently associated with a powerful effect on the growth of neurites of a large number of nanoparticles as topological stimuli. The issue of creating plasmonic ECM-mimetic materials has been repeatedly raised by researchers in studies devoted to light/heat control of neuronal activity [Alghazali, K.M.; Hamzah, R.N.; Nima, Z.A.; Steiner, R.; Dhar, M.; Anderson, D.E.; Hayar, A.; Griffin, R.J.; Biris, A.S. Plasmonic Nanofactors as Switchable Devices to Promote or Inhibit Neuronal Activity and Function. Nanomaterials 2019, 9, 1029. https://doi.org/10.3390/nano9071029; Jung, S.; Harris, N.; Niyonshuti, I.I.; Jenkins, S. V.; Hayar, A.M.; Watanabe, F.; Jamshidi-Parsian, A.; Chen, J.; Borrelli, M.J.; Griffin, R.J. Photothermal Response Induced by Nanocage-Coated Artificial Extracellular Matrix Promotes Neural Stem Cell Differentiation. Nanomaterials 2021, 11, 1216. https://doi.org/10.3390/nano11051216]. Researchers are also wondering if the heat generated by activated surface plasmon structures actually has a stimulating/inhibiting effect on neurons or is it due to other factors, such as the direct action of light, such as in photobiomodulation based on the use of low-level laser therapy or surface properties. In this work, we tried to separate the effects of NIR stimulation and topology-mediated effects on cell behavior. In addition, we measured the change in intracellular temperature during NIR irradiation and recorded an increase in temperature during the activation of plasmonic nanoparticles under exposure to light and the absence of a significant temperature increase in their absence in control samples and on unmodified nanofibers. Temperature data correlate with biological effects observed in cells (growth of processes and differentiation), suggesting that these effects are associated with photo-induced heating. We have expanded the scientific discussion and made corresponding changes into the Manuscript (p. 22, 23).
Point 7. Here are also some minor but also important comments/suggestions:
On line 236, 424, 425, there are some non-English words and characters. There seem to be more throughout the manuscript, which needs correction and consistency.
Response 7. We have carefully examined the text of the manuscript and corrected several grammatical and stylistic errors.
Point 8. On line 424, reference was not completed properly.
Response 8. Apparently the Reviewer had in mind line 724. We have made corrections and have fully indicated the pages.

Reviewer 3 Report
The paper include a rather simple constatation of many views of different-type materials. I have doubts, regarding to real scientific value of these-type "database". In general, scientific discussion, including practical point of views and potential applications of results, should be deeper prepared.
Author Response
Point 1. The paper include a rather simple constatation of many views of different-type materials. I have doubts, regarding to real scientific value of these-type "database". In general, scientific discussion, including practical point of views and potential applications of results, should be deeper prepared.
Response 1. We apologize for not making our scientific discussion clear. The Manuscript has been substantially revised taking into account all the issues raised by Reviewers and Editor. Clarifications have been added regarding the choice of fiber material and the characteristics of the thermoplasmonic nanoparticles used. We also provide data on the magnitude of intracellular heating for different cell types. A discussion of the significance of the biological effect due to photothermal exposure is presented. We have expanded the scientific discussion and introduced potential applications of electrospun composite nanofibers as a promising material for artificial neural conduits.
Round 2
Reviewer 3 Report
The authors made some changes to the text. I appreciate that they put in their effort to improve the manuscript. However, the overall structure of the manuscript still remains as characterized in my original review. In my opinion, based on the research material collected by the Authors, it would be advisable to write a completely new article that would show a structure typical of a scientific work and not a simple set of statements of facts. Unfortunately, I regret to admit that I still do not consider this article in its current form suitable for further evaluation.
Author Response
Response 1. We are grateful for the comments about the structure of the article and repeated statements of facts in a discussion. Taking them into account, we have shortened the text of the Discussion by removing repeated references to the results section, while trying to maintain the logic of the narrative. Additional clarifications have also been added at the request of the Academic Editor. We believe that, taking into account the changes made, the scientific discussion and the practical significance of the results obtained are presented in accordance with the requirements of the journal.